# Analysis of Nanotoxicity with Integrated Omics and Mechanobiology

**DOI:** 10.3390/nano11092385

**Published:** 2021-09-13

**Authors:** Tae Hwan Shin, Saraswathy Nithiyanandam, Da Yeon Lee, Do Hyeon Kwon, Ji Su Hwang, Seok Gi Kim, Yong Eun Jang, Shaherin Basith, Sungsu Park, Jung-Soon Mo, Gwang Lee

**Affiliations:** 1Department of Physiology, Ajou University School of Medicine, Suwon 16499, Korea; catholicon@ajou.ac.kr (T.H.S.); ekdus93@ajou.ac.kr (D.Y.L.); shaherinb@gmail.com (S.B.); 2Department of Molecular Science and Technology, Ajou University, Suwon 16499, Korea; saraswathy.nithiyanandham@gmail.com (S.N.); dohyeon326@gmail.com (D.H.K.); 456547@naver.com (J.S.H.); rlatjrrl9977@naver.com (S.G.K.); 3Department of Biological Sciences, Ajou University, Suwon 16499, Korea; jys613k@gmail.com; 4School of Mechanical Engineering, Sungkyunkwan University, Suwon 16419, Korea; nanopark@skku.edu; 5Institute for Medical Sciences, Ajou University School of Medicine, Suwon 16499, Korea

**Keywords:** integrated omics, machine learning, mechanobiology, nanoparticle, nanotoxicity

## Abstract

Nanoparticles (NPs) in biomedical applications have benefits owing to their small size. However, their intricate and sensitive nature makes an evaluation of the adverse effects of NPs on health necessary and challenging. Since there are limitations to conventional toxicological methods and omics analyses provide a more comprehensive molecular profiling of multifactorial biological systems, omics approaches are necessary to evaluate nanotoxicity. Compared to a single omics layer, integrated omics across multiple omics layers provides more sensitive and comprehensive details on NP-induced toxicity based on network integration analysis. As multi-omics data are heterogeneous and massive, computational methods such as machine learning (ML) have been applied for investigating correlation among each omics. This integration of omics and ML approaches will be helpful for analyzing nanotoxicity. To that end, mechanobiology has been applied for evaluating the biophysical changes in NPs by measuring the traction force and rigidity sensing in NP-treated cells using a sub-elastomeric pillar. Therefore, integrated omics approaches are suitable for elucidating mechanobiological effects exerted by NPs. These technologies will be valuable for expanding the safety evaluations of NPs. Here, we review the integration of omics, ML, and mechanobiology for evaluating nanotoxicity.

## 1. Introduction

Rapidly growing reports on breakthrough applications of NPs have allowed the recognition of their potential uses in biomedical fields [1,2,3,4]. NPs, which have at least one dimension of 100 nm or less, show modifications in the physiochemical properties depending on their size and large surface area to volume ratio as well as changes in the uptake levels and interactions with the human body through inhalation, penetration, and ingestion [1,5]. However, there are concerns regarding their toxicity and safety owing to the adverse biological effects and widespread solicitude of the possible negative impacts of NPs [6,7,8,9,10]. In addition, there are several limitations involved in the evaluation of nanotoxicity with traditional approaches owing to the intricate size-dependent changes and absence of a standard protocol for the assessment of nanotoxicity. Recently, systems toxicology, i.e., the integration of conventional toxicological methods with a comprehensive analysis of in-depth networks and multiple levels of molecules, has been applied for the assessment of nanotoxicity [11]. In particular, omics, which is the core technology in systems toxicology, has been utilized. Therefore, advanced omics approaches have been necessitated for the evaluation of the nanotoxicity patterns caused by NPs.

Omics can be defined as the global assessment of biological molecules (http://omics.org/, access data: 1 February 2021) for understanding their complete genetic or molecular profiles. Recent advances in high-throughput technologies have enabled multi-level omics studies and omnidirectional novel findings in biomedical research [12,13]. Thus, omics approaches are useful in addressing the complexity of biological systems by correcting a full data set. Nowadays, the rapid development of nanotechnology and production of NPs has raised a major issue regarding the potential nanotoxicity [14,15]. Single omics approaches, such as genomics, epigenomics, miRNomics, transcriptomics, proteomics, phosphoproteomics, and metabolomics, are used in the analysis of nanotoxicity [16,17,18,19,20]. However, there are limitations to using a single omics layer because of the limited insights into the interconnected molecular pathways and the complex biological events in cells and organisms [21,22].

Even though single omics approaches are more helpful for the in-depth analysis of biological phenotypes than conventional approaches, the integration of several omics approaches can allow a more comprehensive analysis [21,23]. Thus, the concept of integrated omics was introduced by Dr. Hood in 2003, who suggested a systems biology approach for the integration of different omics data [24]. The conventional method for the assessment of NPs could not detect the actual nanotoxicity and overlooked the mild cytotoxicity in NPs-exposed cells [25,26,27]. However, analysis with systems toxicology provided more sensitive and accurate cytotoxic information in a trans-omics network than traditional in vitro and in vivo and single omics analysis [18,21,28]. Therefore, analysis with systems toxicology, including integrated omics approaches, will be helpful for the precise assessment of nanotoxicity. Given that big data are used for integrated multi-omics, advanced analysis of discrimination methods such as machine learning (ML) has been introduced for their classification [29]. For the precise assessment of nanotoxicity, accurate multi-omics data (big data) and the selection of a proper ML algorithm are crucial.

Mechanobiology is the study of how the physical forces influence cells and tissues and by what means these forces control the behavior of cells. For example, the assessment of cellular force by elastomeric pillar arrays is considered a useful tool for measuring the biophysical aspects of biological phenotypes [30,31]. This approach contributed to the quantitative analyses of the mechanobiological effects in NP-treated cells. Recently, a combination of elastomeric pillar arrays and integrated omics analysis was used for analyzing the mechanisms of mechanobiological effects in NP-treated cells [32,33]. Rather than conventional methods, this approach allowed the evaluation of even mild nanotoxicity and the sensitive evaluation of biophysical conditions. In this paper, we review the four following sections: (i) omics approaches for the assessment of nanotoxicity; (ii) integrated omics for the analysis of nanotoxicity; (iii) ML for the integration of omics; and (iv) mechanobiology for the analysis of nanotoxicity.

## 2. Omics Approaches for the Assessment of Nanotoxicity

Omics approaches provide a more comprehensive understanding of the biological events triggered by NPs than conventional toxicological methods. Importantly, omics data aid in discovering the mechanisms of the biological processes, such as molecular interactions, cellular responses, tissue/organ changes, and organism responses [34]. In addition, they elucidate the complexity of biological functions through statistical analyses and interpretations of bioinformatics. Recently, omics studies of NP-induced biological changes have been increasing in number [16,17,18,19,20]. Despite the high potential of targeting efficiency and the fascinating physicochemical properties of NPs, their adverse effects on cells, tissues, and organs remain a major challenge. Advanced technologies aid in the evaluation of toxicity, cell damage, and other signaling responses caused by NPs, while the traditional methods have limitations for the safety assessment. Therefore, the development of high-throughput technologies for generating various omics data is needed. In this section, we review multi-omics approaches, such as transcriptomics, including RNA and microRNA, proteomics, phosphoproteomics, and metabolomics, for the assessment of nanotoxicity.

### 2.1. Transcriptomics

Transcriptome comprises all the coding and non-coding RNA transcripts, including microRNAs (miRNAs) and long non-coding RNAs (lncRNAs), within the cell. Transcriptomic analysis has been performed for a superior understanding of gene expression patterns and cellular mechanisms. In particular, microRNAs have drawn much attention as new candidate molecules for transcriptional analyses. Transcriptome has been analyzed commonly using two methods, DNA-microarray and Next-generation sequencing (NGS), which are the most recently developed techniques for analyzing gene expression levels [35,36]. Due to high-throughput data generation, transcriptomics has been applied to evaluate the nanotoxicity in cells treated with NPs, such as CuO NPs, nano-Ag (nAg), nano-TiO_2_ (nTiO_2_), nano-ZnO (nZnO), CdTe/CdS, quantum dots (QDs), and silica-coated magnetic nanoparticles containing rhodamine B isothiocyanate dye [MNPs@SiO_2_(RITC)] [21,35,37,38,39,40].

Among transcriptome, microRNA, which is a small, single-stranded non-coding RNA, is the dominant cellular tool that adjusts the levels of all the classes of RNA in post-transcriptional processes through RNA silencing and target genes [41]. The transcriptome plays a vital role in the modulation of cellular mechanisms, such as cell proliferation, metabolism, apoptosis, morphogenesis, and differentiation [42]. Computational prediction methods are needed to understand the comprehensive biological functions of miRNAs, including target gene prediction, and addressing the problems in miRNA-target genome interactions. With the development of rapid sequencing technologies for microRNA, the microRNA has been employed in the assessment of nanotoxicity by treatment with multi-walled carbon nanotubes (MWCNTs) and graphene oxide (GO) in nematodes [43,44,45].

### 2.2. Proteomics

Proteomics is the large-scale study of proteins, which emerged in the late 1970s, to understand the molecular processes of the multifactorial biological systems associated with cellular metabolism and pathogenesis at the protein level [46]. The assessment and quantification of proteins have been analyzed using two-dimensional (2D) gel electrophoresis based on isoelectric point and mass spectrometry, and liquid chromatography-tandem mass spectrometry (LC-MS/MS) for the identification and quantification of the proteome. In addition, protein arrays (protein chips) have also been used for the large-scale analysis of protein interactions and functions [47]. The influence of developing technology on the application of proteomics to nano techniques has continuously evolved [48,49,50]. Proteomics technology covers the evaluation of features, compositions, size, and charges of proteins [51,52,53], and the methods have been used in the assessment of nanotoxicity induced by different types of NPs. NP-induced toxicity has been reported as interference in the cell cycle and damage to the organs by alterations in the expression levels of proteins [54,55,56]. Therefore, proteomics can be one of the approaches used to analyze the cytotoxic mechanisms and side effects induced by NPs, such as the generation of reactive oxygen species (ROS), ER stress, the Fenton reaction, and the oxidation of cellular components.

### 2.3. Phosphoproteomics

Following protein folding in the translation process, proteins undergo post-translational modifications (PTMs), which diversify the proteins’ states and functions, including inactive/active status, protein–protein interactions, and translocation, by the enzymatic covalent addition of functional groups. Thus far, more than 200 types of PTMs have been identified, including phosphorylation, glycosylation, and proteolysis [57]. Among various types of PTMs, protein phosphorylation is particularly involved in a wide range of cellular events. Phosphorylation is modulated by protein kinases and phosphatases and can regulate protein activity and affect interactions between proteins [58]. In contrast to the sole expression analysis of proteins, phosphoproteomics provided clues on what protein and pathway might be activated. MS-based approaches have been used to address the specific quantification of the phosphoproteome, apart from the total protein. Recently, phosphoproteomics has been applied for the assessment of nanotoxicity. For example, Biola-Clier et al. reported that analysis of the phosphoproteome in TiO_2_ NP-treated A549 cells showed that the TiO_2_ NPs altered cellular processes such as apoptosis, cell cycle and DNA damage response, autophagy progress, RNA dynamics, and intracellular transport [20]. In particular, TiO_2_ NPs affected the phosphoproteome related to their upstream regulators. Hence, analysis of phosphoproteome can present a new approach for studying the functions of proteins and signaling pathways in NPs-exposed cells.

### 2.4. Metabolomics

Metabolomics is the large-scale study of metabolites involved in biological processes, such as cellular signaling, repressive or activating effects on enzymes, catalytic activity, interactions with other organisms, energy conversion, and energy metabolism [59,60]. Metabolites are the final products of cellular processes, and among all the types of multi-omics data, phenotypes are the most directly related to metabolomics [21]. Compared to other omics, metabolomics best reflects the phenotypes and metabolic changes rampant in cell masses, biological fluids, and organs [21,61,62]. The efficiency of metabolomics analyses can be elevated by a molecule-targeting approach as well as encountering by using untargeted metabolites. Metabolomics profiling can be achieved through nuclear magnetic resonance (NMR) spectroscopy, gas chromatography-MS (GC-MS), capillary electrophoresis (CE), LC-MS), and LC-MS/MS.

Metabolic profiling using NMR and MS is useful for analyzing dynamic responses, delicate pathophysiological changes in biological systems, and low-molecular-weight metabolites and their intermediates. NMR can detect not only hydrophilic but also hydrophobic samples, such as organic acids, polyols, and alcohols. However, NMR has its limitations; it has 10 to 100 times lower sensitivity than GC-MS or LC-MS [63]. MS is the most widely used technique to measure metabolites; in a sample with single or mixed molecules, using a mass-to-charge ratio (*m/z*) is effective [64]. Metabolic profiling, including amino acids, free fatty acids, organic acids, and polyamines, using MS is a particularly efficient method to evaluate nanotoxicity and provides clues for novel cellular responses and in-depth analysis of the toxicological intracellular pathways affected by NPs, as in the case of MNPs@SiO_2_(RITC) [18,28,32,65]. Even though metabolomics directly reflects cellular phenotypes, using metabolomics without other omics has its limitations since it fails to identify the exact cellular pathways. Thus, integrated omics can be useful to evaluate nanotoxicity, whereby the limitations of each omics approach can be compensated by the others.

## 3. Integrated Omics for the Analysis of Nanotoxicity

The dynamic and complex nature of biological systems is one of the challenges that can be resolved using integrated omics approaches, such as genomics, transcriptomics, proteomics, phosphoproteomics, and metabolomics [66]. This integration of systems biology and systems toxicology was proposed based on the concept of the integration of different types of data, including omics data and cross-disciplinary biology [11,24]. Until now, integrated omics provided a lot of pioneering data for finding biomarkers in nanotoxicity [28], deciphering disease mechanisms [67,68], cancer classification [69], probiotic selection [70], and plant biology [71]. Integrated omics have provided a sensitive and comprehensive analysis of nanotoxicity in the field of systems toxicology by the identification of new pathways and regulator genes/proteins [16,21], and the persistence/recovery mechanism [72]. In addition, integrated omics approaches pave the way to visualizing the stream of information from one omics to other, thus bridging the gap between the genotype and phenotype. The availability of multi-omics data not only revolutionized medicine and biology by constructing novel networks but also directed integrated system-level approaches [73]. In nanotoxicology, metabotranscriptomics was introduced for analyzing subtle biological changes that are undetectable by a conventional toxicological assessment [21,28].

The phenotypes of nanotoxicity vary depending on the dose, composition, shape, and physicochemical properties of nanomaterials [74,75]. Inorganic biocompatible silica [40]; non-toxic degradation, non-immunogenicity, and resistance to protein absorption polyethylene glycol [76]; biodegradable polysaccharides [77,78]; and phospholipids-based liposome [79] have been used as biocompatible nanomaterials and for coating toxic nanomaterials. The biocompatible nanomaterials also trigger cellular toxicity; however, the phenomena occurs at undetectable levels when detected using conventional toxicological assessment methods, such as cell death assay using flow cytometry, MTT, chromosome aberration, in vitro cell cycle analyses and histopathological analysis using hematoxylin and eosin, and clinical assessment (behavior, weight, and growth) using an in vivo mouse model [21,28]. This limitation can be overcome by performing and integrating omics. Omics analysis can reveal holistic biological changes and provide clues for understanding the phenotypic changes in NP-treated cells and organs [80].

Gene and transcript expression in cells or tissues can be analyzed using NGS and also be investigated for gene function, structure, and alternative splicing [81]. However, there are limitations in the interpretation of transcriptome where a mismatch affecting the abundance of mRNAs or proteins is observed. In human cells, for example, only 27% of the protein content can be explained by mRNA abundance, and the remaining are explained by other factors [82]. The basic status of omics research tends to use only single omics to describe specific phenomena. However, single omics analysis has a “blind spot” for understanding complex biological mechanisms due to the disparity between omics data and actual phenotypes [80]. Thus, to compensate for the cons of each single omics, integrated omics is highly recommended.

Integrated omics have provided an unprecedented extent of intracellular cytotoxic phenomena compared to conventional methods for nanotoxicity evaluation. For example, the internalization efficiencies of MNPs@SiO_2_(RITC) were investigated in various cell types, and the MNPs@SiO_2_(RITC) was expected to lead to the development of various applications in cell separation, biological labeling and detection, and drug and delivery [40]. This NP showed no cytotoxicity in traditional in vitro assays, such as the observation of cell morphology with optical imaging, fluorescence-activated cell sorting (FACS) analysis, a cell viability assay, an apoptosis assay, a cell cycle arrest assay, and a chromosome aberration assay [21,27,28]. In addition, this NP showed no abnormal cytotoxicity with traditional in vivo assays, such as pathological analysis with hematoxylin and eosin staining; analysis of the blood–brain barrier (BBB) permeability; health behaviors with growth, body weight, and behaviors; and hematological test with glucose, cholesterol, and creatinine [25]. However, the integration of metabolomics and transcriptomics, i.e., metabotranscriptomics, revealed detailed toxicological mechanisms and evaluated cellular responses triggered by MNPs@SiO_2_(RITC) [18,28,83]. To analyze the phenomena occurring in MNPs@SiO_2_(RITC)-treated cells, a single integrated network of genes and metabolites was constructed using a transcriptome generated from microarray analysis and the amino acid and organic acid metabolomes derived from GC-MS analysis using Ingenuity Pathway Analysis (IPA, http://www.ingenuity.com, access data: 1 July 2021), as shown in Figure 1 and Table 1 based on the previous reports [18,21,28,65]. This integrated network showed that the genes and metabolome were related to three biological functions, including ROS generation, glucose metabolism disorder, and reduced ATP synthesis, in 1.0 µg/µL MNPs@SiO_2_(RITC)-treated cells.

Shin et al. analyzed the impaired glucose metabolism in MNPs@SiO_2_(RITC)-treated HEK293 cells using metabotranscriptomics analysis [18]. Glucose metabolism disorder is closely related to ROS generation, and the dysfunction in glucose uptake in MNPs@SiO_2_(RITC)-treated HEK293 cells was identified as the fundamental factor underlying impaired glucose metabolism. In silico metabotranscriptomics accurately deduced the aforementioned biological changes and indicated the possibility of constructing a model in the future for assessing toxicity using integrated omics analysis.

## 4. ML for the Integration of Omics

Computational approaches and artificial intelligence (AI) have been considered for handling vast amounts of generated big data, such as single or integrated omics data. The integration of omics data has been enhanced by AI approaches such as ML, which are supportive in handling large-scale datasets. ML is an adaptive process that directs automated learning of machines without being programmed explicitly [84]. ML models allow researchers to mine the multi-omics data that hold great promise in unraveling the complex relationships associated with the molecular features [34,73]. ML can be divided into the following three categories: *(a)* supervised, *(b)* unsupervised, and *(c)* reinforcement or semi-supervised. Based on the data resemblance, an unsupervised ML model learns patterns from the unlabeled dataset and groups them accordingly [84]. The contemplated supervised models in the context of ML include the support vector machine (SVM), the K-nearest neighbor (KNN), and the Naïve Bayes method [85,86,87]. These approaches are recommended to be applied for challenging pattern-recognition problems in biological data. In the branch of classification, K-means clustering and Hierarchical Density-Based Spatial Clustering of Applications with Noise (HDBSCAN) are traditional approaches in unsupervised ML, which can handle huge datasets such as multi-omics data to generate globular-shaped tight clustering using less computational time. In addition, deep learning (DL), a subset of ML, can progressively extract comprehensive features from the hidden layers of the integrated-omics data [88,89].

With the fast adoption of high-throughput technologies, such as NGS and deep sequencing, a single experiment is enough to measure the molecular parameters in large-scale data termed as omics, such as genomics, transcriptomics, proteomics, phosphoproteomics, and microRNA-omics [36]. A complete understanding of the NP-induced toxicity and biological changes in cells is highly challenging; thus, advanced omics data and ML are required to bridge the gap between the computational approach (ML model) and biological data (multi-omics data). Moreover, the advancement of ML approaches aids in the development of novel assessment strategies for NP-induced toxicity [90,91]. A nanotoxicity-based ML classification model was constructed for different nanoparticles using literature-based data focused on cell type, cell origin, assay method, NP type, physicochemical properties, and exposure parameters (route, duration, and concentration) [90]. Peng et al. generated a metabolic pathway-based (amino acid, lipid, carbohydrate, energy, nucleotide, and biosynthesis) prediction model using a nanotoxicity database for 33 kinds of NPs in animals, cells, and plants [29]. Oh et al. studied the correlation between physicochemical properties and toxicity quantum dots using an ML regression model with database from 307 publications further identifying that the shell, ligand, and surface modifications, diameter, assay type, and exposure time were closely correlated with toxicity [92].

The workflow of multi-omics integration with ML is shown in Figure 2. All individual omics data can be gathered using high-throughput technologies, including genomics, transcriptomics (mRNA, miRNA), proteomics, phosphoproteomics, and metabolomics. Subsequently, statistical analysis can be performed as a preprocessing step in the integration of omics using ML methods. At the end of the synopsis, supervised and unsupervised methods can be chosen based on the target. In the supervised approach, feature selection and feature extraction play a pivotal role in finding the most relevant data to build the model using the DL approach for the extraction of the input from the data. The outcome is an integrated omics approach that can build novel networks and identify pathways or targeting molecules using the networks.

## 5. Mechanobiology for the Analysis of Nanotoxicity

Many technological platforms and evaluation systems are being developed to study the mechanical changes in cells, which has led to the discovery of biophysical phenotypes and molecular and cellular mechanisms. There are various types of mechanical force measurement techniques, such as traction force and rigidity sensing that can be used for studying how cellular forces are built into biological and functional responses [93,94]. Additionally, to identify the corresponding mechanotransduction signaling pathways, atomic force microscopy (AFM), micropipette aspiration, optical tweezers, magnetic tweezers, and a uniaxial stretcher have been used for measuring mechanical forces [95]. Although mechanobiology can be deemed a good approach for assessing nanotoxicity owing to its ability to analyze the physical changes and forces in the mechanical properties of cells [96], mechanobiology has not been well studied for the assessment of nanotoxicity.

Elastomeric pillar arrays are considered a good mechanobiological tool to assess cellular forces in the nanonewton range as it can calculate the nanometric level of pillar deflection [30,31,97]. Polydimethylsiloxane (PDMS)-based sub-micron elastomeric pillar arrays can be used to analyze the traction force and rigidity sensing of an initial cell contact with a substrate [97,98]. During the initial contact between cells and substrates such as an extracellular matrix (ECM), cells can sense the rigidity of the substrate for adaptation by the regulating biological changes, based on the mechanical properties of the substrate [99,100,101]. Intrinsically, the impairment of rigidity sensing is expressed with changes in cell morphology and focal adhesion status [102], and the directionality of local contraction can be measured with deflection of the neighboring submicron pillars in a PDMS-based micropillar [33]. Recently, the biophysical effects in MNPs@SiO_2_(RITC)-treated cells could be measured quantitatively, in the forms of traction force and the rigidity sensing with elastomeric pillar arrays [32,33]. Therefore, applying mechanobiology for the assessment of the biophysical phenotypes and molecular cellular mechanisms of NPs-treated cells will be useful.

Figure 3 shows the metabotranscriptomic network used for the analysis of changes in traction forces and rigidity sensing in MNPs@SiO_2_(RITC)-treated HEK293 cells [32,33]. Cell shrinkage of MNPs@SiO_2_(RITC)-treated cells is significantly decreased by lipid peroxidation with ROS, and this phenomenon was analyzed using metabotranscriptomics. The toxicological mechanisms and the evaluation of cellular responses activated by MNPs@SiO_2_(RITC) are depicted in Figure 3a [32,33]. The biophysical changes were analyzed using metabotranscriptomic network approaches, and biological functions, such as lipid peroxidation, focal adhesion, and cell movement, were tightly related to each other by changes in the metabolome and transcriptome. In the calculation of the traction force of a pillar, 1.0 µg/µL MNPs@SiO_2_(RITC)-treated cells showed a significant increase in pillar displacement, which was substantially higher than the pillar displacement of the non-treated control cells (Figure 3b). The results showed that cell traction force was affected by MNPs@SiO_2_(RITC) at 1.0 µg/µL concentration. However, for the assessment of the directionality parameter, local contractions of 0.1 and 1.0 µg/µL MNPs@SiO_2_(RITC)-treated cells were significantly reduced compared to the non-treated control, indicating that the analysis of rigidity sensing is more sensitive compared to that of the traction force (Figure 3c). The analysis of the metabotranscriptomic network in MNPs@SiO_2_(RITC)-treated HEK293 cells also showed an increment in lipid peroxidation, impaired focal adhesion, and reduced cell movement, which are related with traction force and rigidity sensing [32,33]. Thus, integrated omics is useful in elucidating the mechanisms of the mechanobiological effects exerted by NPs.

## 6. Summary

We summarized the toxicological evaluations of MNPs@SiO_2_(RITC)-treated cells using integrated omics, ML, and mechanobiology. We expect that these kinds of toxicological evaluations will be applicable for the validation of various nanomaterial-induced nanotoxicity, including lipid and organic macromolecules-based nanoparticles, using high throughput omics analysis, ML-based omics integration, and an assessment of biophysical changes. Furthermore, the reported results in this review were analyzed at one time point, whereas cellular responses are sequentially altered. Thus, inter time omics and mechanobiology analyses are highly recommended for the detailed evaluation of nanotoxicity with respect to early acute response and late chronic response against the nanoparticles. In conclusion, a combination of integrated omics and mechanobiology will be helpful in analyzing biophysical phenotypes and the underlying molecular cellular mechanisms.

Here, we reviewed advanced omics techniques, the integration of omics, ML, and the mechanobiology employed in evaluating nanotoxicity. Although classical procedures are valued as cornerstones of nanotoxicity analyses, they have their limitations in detecting nanotoxicity. The introduction of advanced tools allowed the integration of omics, ML, and mechanobiology, and the technological progress in molecular detection methods has the potential to further improve omics techniques. It is not only the integration of omics with ML, but the integration of mechanobiology with these methods in nanotoxicity research that can contribute to analyzing the delicate mechanobiological effects in NP-treated cells. Therefore, the integration of omics, ML, and mechanobiology might be helpful for conducting a precise safety evaluation for NPs in biomedical research.

## Figures and Tables

**Figure 1 nanomaterials-11-02385-f001:**
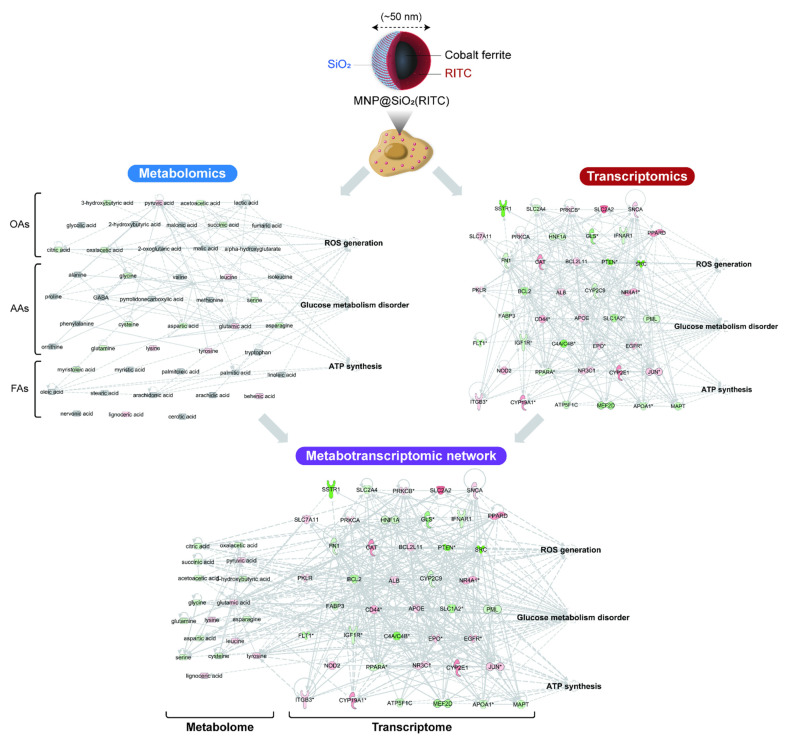
The metabotranscriptomic network and evaluation of ROS generation, glucose metabolism disorder, and ATP synthesis in 1.0 µg/µL MNPs@SiO_2_(RITC)-treated cells. Red and green areas indicate upregulated and downregulated factors, respectively, compared with those of the untreated control group. OAs; organic acids, AAs; amino acids, FAs: fatty acids. The data were reproduced from our previous studies [18,28].

**Figure 2 nanomaterials-11-02385-f002:**
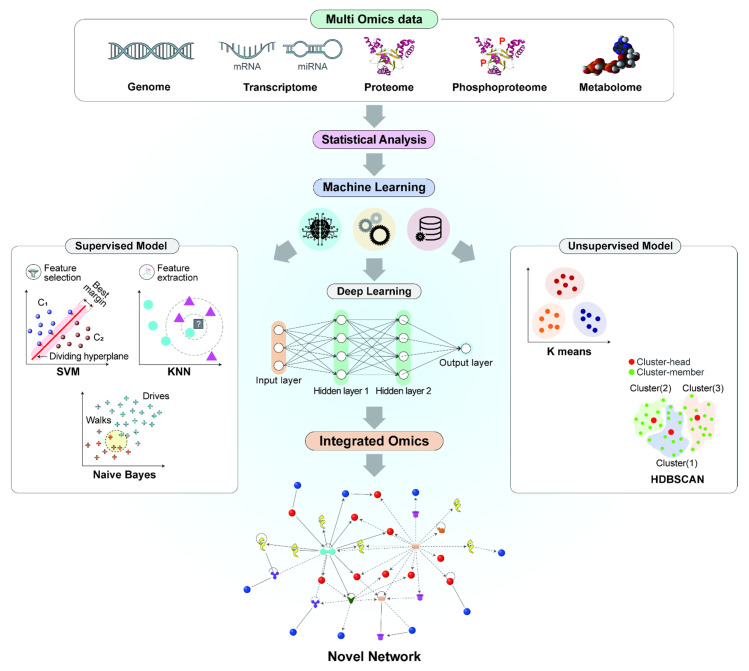
Overview of the ML approaches in the integration of multi-omics data. Multi-omics data could be integrated using both statistical and ML methods, where statistical approaches are used as a preprocessing step. ML approaches are classified into supervised and unsupervised algorithms, which could be utilized in the integration of omics to reduce the data dimensionality for building novel networks and pathways. SVM: support vector machine; KNN: K-nearest neighbor; HDBSCAN: hierarchical density-based spatial clustering of applications with noise.

**Figure 3 nanomaterials-11-02385-f003:**
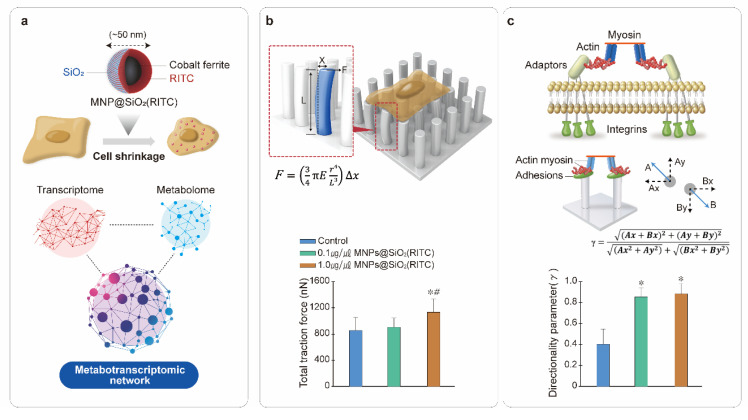
Analysis of the mechanism network of NP-treated cells using metabotranscriptomics (**a**), traction force (**b**), and rigidity sensing (**c**). Metabotranscriptomic network was analyzed in cells treated with MNPs@SiO_2_(RITC) in previous reports [21,28]. Submicron elastomeric micropillars were used to measure the traction force (*F*) and directionality parameter (γ) in MNPs@SiO_2_(RITC)-treated cells. The data were reproduced from our previous study [32,33].

**Table 1 nanomaterials-11-02385-t001:** Ingenuity Pathway Analysis-based profiles of ROS generation, glucose metabolism disorder, and ATP synthesis related factors of HEK293 cells treated with MNPs@SiO_2_(RITC).

Entrez Gene Name	Symbol	ID ^b^	Location	Signal Fold Change ^a^	
				0.1 μg/μL	1.0 μg/μL
albumin	ALB	1565228_s_at	Extracellular Space	5.50	4.16
apolipoprotein A1	APOA1	217073_x_at	Extracellular Space	−3.75	−1.53
apolipoprotein E	APOE	203382_s_at	Extracellular Space	5.13	3.38
ATP synthase F1 subunit gamma	ATP5F1C	214132_at	Cytoplasm	−6.06	−1.28
BCL2 apoptosis regulator	BCL2	203684_s_at	Cytoplasm	−10.46	−7.26
BCL2 like 11	BCL2L11	208536_s_at	Cytoplasm	6.37	1.70
catalase	CAT	215573_at	Cytoplasm	17.41	13.38
CD44 molecule (Indian blood group)	CD44	234411_x_at	Plasma Membrane	11.38	−4.89
complement C4A (Rodgers blood group)	C4A/C4B	208451_s_at	Extracellular Space	−20.71	9.05
cytochrome P450 family 19 subfamily A member 1	CYP19A1	1554296_at	Cytoplasm	15.56	−17.51
cytochrome P450 family 2 subfamily C member 9	CYP2C9	214419_s_at	Cytoplasm	−3.60	1.19
cytochrome P450 family 2 subfamily E member 1	CYP2E1	209976_s_at	Cytoplasm	18.60	3.69
epidermal growth factor receptor	EGFR	1565484_x_at	Plasma Membrane	9.62	9.52
erythropoietin	EPO	207257_at	Extracellular Space	8.02	8.34
fatty acid binding protein 3	FABP3	214285_at	Cytoplasm	−3.13	−4.84
fibronectin 1	FN1	214702_at	Extracellular Space	−4.96	1.89
fms related receptor tyrosine kinase 1	FLT1	210287_s_at	Plasma Membrane	−10.50	−3.85
glutaminase	GLS	203158_s_at	Cytoplasm	−14.49	−2.26
HNF1 homeobox A	HNF1A	216930_at	Nucleus	−4.21	1.60
insulin like growth factor 1 receptor	IGF1R	243358_at	Plasma Membrane	−4.50	−1.44
integrin subunit beta 3	ITGB3	215240_at	Plasma Membrane	5.79	5.93
interferon alpha and beta receptor subunit 1	IFNAR1	204191_at	Plasma Membrane	−5.46	1.37
Jun proto-oncogene, AP-1 transcription factor subunit	JUN	201465_s_at	Nucleus	7.34	1.43
microtubule associated protein tau	MAPT	203930_s_at	Plasma Membrane	−8.28	−1.43
myocyte enhancer factor 2D	MEF2D	225641_at	Nucleus	−11.89	−3.92
nuclear receptor subfamily 3 group C member 1	NR3C1	232431_at	Nucleus	4.01	−1.09
nuclear receptor subfamily 4 group A member 1	NR4A1	211143_x_at	Nucleus	11.14	4.23
nucleotide binding oligomerization domain containing 2	NOD2	220066_at	Cytoplasm	9.56	7.03
peroxisome proliferator activated receptor alpha	PPARA	1560981_a_at	Nucleus	−4.81	6.60
peroxisome proliferator activated receptor delta	PPARD	210636_at	Nucleus	15.79	7.90
phosphatase and tensin homolog	PTEN	240964_at	Cytoplasm	−24.40	4.66
promyelocytic leukemia	PML	239582_at	Nucleus	−5.15	1.31
protein kinase C alpha	PRKCA	215195_at	Cytoplasm	3.53	2.76
protein kinase C beta	PRKCB	227824_at	Cytoplasm	6.41	−10.37
pyruvate kinase L/R	PKLR	207858_s_at	Cytoplasm	3.38	1.11
solute carrier family 1 member 2	SLC1A2	217055_x_at	Plasma Membrane	−11.34	−8.32
solute carrier family 2 member 2	SLC2A2	206535_at	Plasma Membrane	26.34	1.90
solute carrier family 2 member 4	SLC2A4	206603_at	Plasma Membrane	−5.42	3.00
solute carrier family 7 member 11	SLC7A11	217678_at	Plasma Membrane	5.61	2.00
somatostatin receptor 1	SSTR1	235591_at	Plasma Membrane	−29.18	−1.32
SRC proto-oncogene, non-receptor tyrosine kinase	SRC	1558210_at	Cytoplasm	−33.92	−12.56
synuclein alpha	SNCA	236081_at	Cytoplasm	3.22	−1.89
	3-hydroxybutyric acid	300-85-6	Other	−1.49	−1.39
	acetoacetic acid	541-50-4	Other	−1.40	−1.40
	citric acid	77-92-9	Other	−1.39	−1.27
	glycine	56-40-6	Other	−1.53	−1.07
	asparagine	70-47-3	Other	−1.21	1.01
	aspartic acid	56-84-8	Other	−1.22	−1.12
	cysteine	52-90-4	Other	−1.24	1.09
	glutamic acid	56-86-0	Other	1.50	1.24
	glutamine	56-85-9	Other	−1.89	−1.51
	leucine	61-90-5	Other	1.38	1.28
	lysine	56-87-1	Other	1.91	1.25
	serine	56-45-1	Other	−1.97	2.12
	tyrosine	60-18-4	Other	1.21	−1.09
	lignoceric acid	557-59-5	Other	1.33	1.21
	oxalacetic acid	328-42-7	Other	−1.47	−1.32
	pyruvic acid	127-17-3	Other	1.41	−6.13
	succinic acid	110-15-6	Other	−2.28	−1.21

^a^ Fold change of normalized signal in MNPs@SiO_2_(RITC)-treated group relative to corresponding control group. ^b^ ID for identifying the factors: Affymetrix probe ID for genes and CAS Registry Number or PubChem CID for metabolite. The data were reproduced from our previous studies, Copyright © 2021, American Chemical Society and Copyright © 2021, Shin et al. [18,28].

## Data Availability

The data presented in this study are openly available in [18,32,33] and the additional data supporting the findings of this study are available from the corresponding author, upon reasonable request.

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
