# Peer review of "Analysis of Nanotoxicity with Integrated Omics and Mechanobiology"

_nanomaterials, 2021, doi:10.3390/nano11092385_

Round 1
Reviewer 1 Report
Authors summarized the combinational approach of integrated omics with ML and mechanobiology for evaluating nanotoxicity. This review is impressive for readers in the field of nanomedicine and the approach is expected as a gold standard method for evaluating nanotoxicity.
Though the manuscript, authors introduce the evaluation of inorganic compounds-based nanoparticles. As you know, lipids and organic macromolecules-based nanoparticles is expected as their application to biomedical fields. Can authors introduce the toxicological evaluation of these nanoparticles using integrated omics with ML and mechanobiology? Although various kinds of omics data may be easily integrated at one time point, cellular responses against nanoparticles are sequentially altered. In such case, how are omics data integrated between different time points?
Author Response
Reviewer 1
Authors summarized the combinational approach of integrated omics with ML and mechanobiology for evaluating nanotoxicity. This review is impressive for readers in the field of nanomedicine and the approach is expected as a gold standard method for evaluating nanotoxicity.
Though the manuscript, authors introduce the evaluation of inorganic compounds-based nanoparticles. As you know, lipids and organic macromolecules-based nanoparticles is expected as their application to biomedical fields. Can authors introduce the toxicological evaluation of these nanoparticles using integrated omics with ML and mechanobiology?
à We agree with your suggestions. We added discussion about the toxicological evaluation of nanoparticles for lipids and organic macromolecules using integrated omics with ML and mechanobiology in line 380 page 12, under “Summary” section.
We summarized toxicological evaluations of MNPs@SiO2(RITC)-treated cells using integrated omics, ML, and mechanobiology. We expect that these kinds of toxicological evaluations will be applicable for the validation of various nanomaterial-induced nanotoxicity, including lipid and organic macromolecules-based nanoparticles, using high throughput omics analysis, ML-based omics integration, and assessment of biophysical changes. Furthermore, the reported results in this review were analyzed at one time point, whereas cellular responses are sequentially altered. Thus, inter time omics and mechanobiology analyses are highly recommended for detailed evaluation of nanotoxicity with respect to early acute response and late chronic response against the nanoparticles. In conclusion, a combination of integrated omics and mechanobiology will be helpful in analyzing biophysical phenotypes and the underlying molecular cellular mechanisms.
Although various kinds of omics data may be easily integrated at one time point, cellular responses against nanoparticles are sequentially altered. In such case, how are omics data integrated between different time points?
à Due to the limitation of resources and data processing methods, omics data at one time point have been reported so far. We also agree with the sequential alteration of cellular responses against nanoparticles. Thus, we added few points about the importance of omics analysis for inter time changes in the nanoparticle treated cells with respect to early acute response and late chronic response against nanoparticles in line 384, page 12.
Furthermore, the reported results in this review were analyzed at one time point, whereas cellular responses are sequentially altered. Thus, inter time omics and mechanobiology analyses are highly recommended for detailed evaluation of nanotoxicity with respect to early acute response and late chronic response against the nanoparticles.

Reviewer 2 Report
Here is my opinion on the manuscript (type – review) „Analysis of nanotoxicity with integrated omics and mechanobiology“ by T. H. Shin et al. (nanomaterials-1336465-peer-review-v1).
The manuscript is (generally) correctly organized, and an overview is given of evaluating nanotoxicity of nanoparticles based on the integration of omics, ML and mechanobiology methods. However, some details need to be improved, as I explain and specify below.
1)
In general, I think that certain parts of the review of the topic (area) are presented mainly in a statement (assertive) mode, without sufficient critical reviews and essential comparisons of results and methods in/between references. Eg. in the sentence (lines 208,209): "Moreover, the advancement of ML approaches aids in the development of novel assessment strategies for NP-induced toxicity [93,94]."
->details and an explanation of how each paper contributed to the improvement of NP-toxicity research using ML approaches should/must be given (naturally – for a scientific review).
I would suggest that the authors work on this and correct other parts that are generally declarative - in order to give the key specificities and contributions of the individual references that they cite in this review paper.
2)
In addition, in my opinion, a little more detail should be given about each reference (or 2, eventually 3 references together) that is cited in all cases where more than 2-3 references at once in one sentence are cited by one general statement. Examples of this are best seen in the first part of Chapter 3 in lines 193-198.
"Until now, integrated omics provided a lot of pioneering data on various physiological conditions and abnormal processes, including pathological conditions [28,66-71], and can be used for understanding complex biological changes [66,67,72-74]. Integrated omics have provided more sensitive and comprehensive analysis of nanotoxicity than conventional methods in the field of systems toxicology [16,21,50,75]. "
Namely, 13 references were cited here and a little was said about each of them particularly (and about the topics they deal with and the research results and news that they brought). Additionally, all the statements made in this section (in these two sentences) are quite general. In such cases, the specific contribution of each (or possibly two together) reference must be specified.
This is true in general – i.e. for all similar parts of the text where more than 2 (or possibly three) references are invoked in one sentence by a general quotation. This needs to be corrected/improved by the authors.
3)
Just a small suggestion - maybe it would be better to write "a more sensitive" in "Integrated omics have provided more sensitive and comprehensive analysis ..."
4)
One example that can help authors improve manuscripts in sentences where citations of more than one reference are present - according to my suggestions above - on the example of the sentence in Lines 206-209:
"Biocompatible nanomaterials, including silica, polyethylene glycol, and polysaccharides [40,79-82], trigger cellular toxicity; however, the phenomena occur at undetectable levels when detected by conventional toxicological assessment methods [21,28]."
In part:
"Biocompatible nanomaterials, including silica, polyethylene glycol, and polysaccharides [40,79-82]"
if silica is related to ref. 40, polyethylene glycol to ref. 79 and polysaccharides to ref. 80.81, this part can be improved to:
" Biocompatible nanomaterials, including silica [40], polyethylene glycol [79], and polysaccharides [80,81] .... "
Also, even a little more specific detail (s) (i.e. one-two-three words more) can be given for describing specifics of each reference.
Author Response
Reviewer 2
Here is my opinion on the manuscript (type – review) „Analysis of nanotoxicity with integrated omics and mechanobiology“ by T. H. Shin et al. (nanomaterials-1336465-peer-review-v1).
The manuscript is (generally) correctly organized, and an overview is given of evaluating nanotoxicity of nanoparticles based on the integration of omics, ML and mechanobiology methods. However, some details need to be improved, as I explain and specify below.
1) In general, I think that certain parts of the review of the topic (area) are presented mainly in a statement (assertive) mode, without sufficient critical reviews and essential comparisons of results and methods in/between references. Eg. in the sentence (lines 208,209): "Moreover, the advancement of ML approaches aids in the development of novel assessment strategies for NP-induced toxicity [93,94]."
->details and an explanation of how each paper contributed to the improvement of NP-toxicity research using ML approaches should/must be given (naturally – for a scientific review).
I would suggest that the authors work on this and correct other parts that are generally declarative - in order to give the key specificities and contributions of the individual references that they cite in this review paper.
à We agree with your suggestions. We added both the details and explanation of each paper.
Line 214 page 5
however, the phenomena occurs at undetectable levels when detected by conventional toxicological assessment methods, such as cell death assay using flow cytometry, MTT, chromosome aberration, in vitro cell cycle analyses and histopathological analysis using hematoxylin and eosin, and clinical assessment (behavior, weight, and growth) using in vivo mouse model [21,28].
Line 298 page 9
With the fast adoption of high-throughput technologies, such as NGS and deep sequencing, a single experiment is enough to measure molecular parameters in a large-scale data termed as omics, such as genomics, transcriptomics, proteomics, phosphoproteomics, and microRNA-omics [36]. Complete understanding of the NP-induced toxicity and biological changes in cells is highly challenging; thus, advanced omics data and ML are required to bridge the gap between the computational approach (ML model) and biological data (multi-omics data). Moreover, the advancement of ML approaches aids in the development of novel assessment strategies for NP-induced toxicity [90,91]. Nanotoxicity-based ML classification model was constructed for different nanoparticles using literature-based data focused on cell type, cell origin, assay method, NP type, physicochemical properties, and exposure parameters (route, duration, and concentration) [90]. Peng et al. generated metabolic pathway-based (amino acid, lipid, carbohydrate, energy, nucleotide, and biosynthesis) prediction model using nanotoxicity database for 33 kinds of NPs in animals, cells, and plants [29]. Oh et al. studied correlation between physicochemical properties and toxicity quantum dots using ML regression model with database from 307 publications further identifying that the shell, ligand and surface modifications, diameter, assay type and exposure time were closely correlated with toxicity [92].
Additional References
- Peng, T.; Wei, C.; Yu, F.; Xu, J.; Zhou, Q.; Shi, T.; Hu, X. Predicting nanotoxicity by an integrated machine learning and metabolomics approach. Environ Pollut 2020, 267, 115434, doi:10.1016/j.envpol.2020.115434.
- Furxhi, I.; Murphy, F. Predicting In Vitro Neurotoxicity Induced by Nanoparticles Using Machine Learning. Int J Mol Sci 2020, 21, 5280, doi:10.3390/ijms21155280.
- Winkler, D.A. Role of Artificial Intelligence and Machine Learning in Nanosafety. Small 2020, 16, 2001883, doi:10.1002/smll.202001883.
- Oh, E.; Liu, R.; Nel, A.; Gemill, K.B.; Bilal, M.; Cohen, Y.; Medintz, I.L. Meta-analysis of cellular toxicity for cadmium-containing quantum dots. Nature Nanotechnology 2016, 11, 479-486, doi:10.1038/nnano.2015.338.
2) In addition, in my opinion, a little more detail should be given about each reference (or 2, eventually 3 references together) that is cited in all cases where more than 2-3 references at once in one sentence are cited by one general statement. Examples of this are best seen in the first part of Chapter 3 in lines 193-198.
"Until now, integrated omics provided a lot of pioneering data on various physiological conditions and abnormal processes, including pathological conditions [28,66-71], and can be used for understanding complex biological changes [66,67,72-74]. Integrated omics have provided more sensitive and comprehensive analysis of nanotoxicity than conventional methods in the field of systems toxicology [16,21,50,75]. "
Namely, 13 references were cited here and a little was said about each of them particularly (and about the topics they deal with and the research results and news that they brought). Additionally, all the statements made in this section (in these two sentences) are quite general. In such cases, the specific contribution of each (or possibly two together) reference must be specified.
This is true in general – i.e. for all similar parts of the text where more than 2 (or possibly three) references are invoked in one sentence by a general quotation. This needs to be corrected/improved by the authors.
à As per reviewer’s suggestions, the general quotations were specified and irrelevant references were removed.
Line 180 page 4
Metabolic profiling using NMR and MS is useful for analyzing dynamic responses, delicate pathophysiological changes in biological systems, and low-molecular-weight metabolites and their intermediates. NMR can detect not only hydrophilic but also hydrophobic samples, such as organic acids, polyols, and alcohols. However, NMR has its limitations; it has 10 to 100 times lower sensitivity than GC-MS or LC-MS [65]. MS is the most widely used technique to measure metabolites; in a sample with single or mixed molecules, using mass-to-charge ratio (m/z) is effective [66]. Metabolic profiling, including amino acids, free fatty acids, organic acids, and polyamines, using MS is a particularly efficient method to evaluate nanotoxicity and provides clues for novel cellular responses and in-depth analysis of toxicological intracellular pathways affected by NPs as in the case of MNPs@SiO2(RITC) [20,30,34,67]. Even though metabolomics directly reflects cellular phenotypes, using metabolomics without other omics has its limitations since it fails to identify the exact cellular pathways. Thus, integrated omics can be useful to evaluate nanotoxicity, whereby the limitations of each omics approach can be compensated by the others.
Line 195 page 4
The dynamic and complex nature of biological systems is one of the challenges that can be resolved using integrated omics approaches, such as genomics, transcriptomics, proteomics, phosphoproteomics, and metabolomics [66]. This integration of systems biology and systems toxicology was proposed based on the concept of integration of different types of data, including omics data and cross-disciplinary biology [11,24]. Until now, integrated omics provided a lot of pioneering data for finding biomarkers in nanotoxicity [28], deciphering disease mechanism [67,68], cancer classification [69], probiotic selection [70], and plant biology [71]. Integrated omics have provided a sensitive and comprehensive analysis of nanotoxicity in the field of systems toxicology by identification of new pathways and regulator genes/proteins [16,21], and persistence/recovery mechanism [72]. In addition, integrated omics approaches pave the way to visualizing the stream of information from one omics to other, thus bridging the gap between the genotype and phenotype. The availability of multi-omics data not only revolutionized medicine and biology by constructing novel networks but also directed integrated system-level approaches [73]. In nanotoxicology, metabotranscriptomics was introduced for analyzing subtle biological changes that are undetectable by conventional toxicological assessment [21,28].
3) Just a small suggestion - maybe it would be better to write "a more sensitive" in "Integrated omics have provided more sensitive and comprehensive analysis ..."
à We corrected the part as reviewer’s suggestion.
Line 197 page 4
Integrated omics have provided a sensitive and comprehensive analysis of nanotoxicity in the field of systems toxicology by identification of new pathways and regulator genes/proteins [16,21], and persistence/recovery mechanism [72].
4) One example that can help authors improve manuscripts in sentences where citations of more than one reference are present - according to my suggestions above - on the example of the sentence in Lines 206-209:
"Biocompatible nanomaterials, including silica, polyethylene glycol, and polysaccharides [40,79-82], trigger cellular toxicity; however, the phenomena occur at undetectable levels when detected by conventional toxicological assessment methods [21,28]."
In part:
"Biocompatible nanomaterials, including silica, polyethylene glycol, and polysaccharides [40,79-82]"
if silica is related to ref. 40, polyethylene glycol to ref. 79 and polysaccharides to ref. 80.81, this part can be improved to:
" Biocompatible nanomaterials, including silica [40], polyethylene glycol [79], and polysaccharides [80,81] .... "
Also, even a little more specific detail (s) (i.e. one-two-three words more) can be given for describing specifics of each reference.
à We separated the part and added specific details as your suggestion.
Line 208 page 5
Phenotypes of nanotoxicity vary depending on dose, composition, shape, and physicochemical properties of nanomaterials [74,75]. Inorganic biocompatible silica [40]; non-toxic degradation, non-immunogenicity, and resistance to protein absorption polyethylene glycol [76]; biodegradable polysaccharides [77,78]; and phospholipids-based liposome [79] have been used as biocompatible nanomaterials and for coating toxic nanomaterials. The biocompatible nanomaterials also trigger cellular toxicity; however, the phenomena occurs at undetectable levels when detected by conventional toxicological assessment methods, such as cell death assay using flow cytometry, MTT, chromosome aberration, in vitro cell cycle analyses and histopathological analysis using hematoxylin and eosin, and clinical assessment (behavior, weight, and growth) using in vivo mouse model [21,28]. This limitation can be overcome by performing and integrating omics. Omics analysis can reveal holistic biological changes and provide clues for understanding the phenotypic changes in NP-treated cells and organs [80].

Round 2
Reviewer 2 Report
-.